

# Long non-coding RNA L13Rik promotes high glucose-induced mesangial cell hypertrophy and matrix protein expression by regulating miR-2861/CDKN1B axis

Linlin Sun, Miao Ding, Fuhua Chen, Dingyu Zhu and Xinmiao Xie

Nephrology, Tongren Hospital, Shanghai Jiao Tong University School of Medicine, Shanghai, China

Corresponding author
Linlin Sun, llsunzj@sina.com

## ABSTRACT

**Background:** Diabetic nephropathy (DN) is a frequent microvascular complication of diabetes. Glomerular mesangial cell (MC) hypertrophy occurs at the initial phase of DN and plays a critical role in the pathogenesis of DN. Given the role of long non coding RNA (lncRNA) in regulating MC hypertrophy and extracellular matrix (ECM) accumulation, our aim was to identify functional lncRNAs during MC hypertrophy.

**Methods:** Here, an lncRNA, C920021L13Rik (L13Rik for short), was identified to be up-regulated in DN progression. The expression of L13Rik in DN patients and diabetic mice was assessed using quantitative real-time PCR (qRT-PCR), and the function of L13Rik in regulating HG-induced MC hypertrophy and ECM accumulation was assessed through flow cytometry and western blotting analysis.

**Results:** The L13Rik levels were significantly increased while the miR-2861 levels were decreased in the peripheral blood of DN patients, the renal tissues of diabetic mice, and HG-treated MCs. Functionally, both L13Rik depletion and miR-2861 overexpression effectively reduced HG-induced cell hypertrophy and ECM accumulation. Mechanistically, L13Rik functioned as a competing endogenous RNA (ceRNA) to sponge miR-2861, resulting in the de-repression of cyclin-dependent kinase inhibitor 1B (CDKN1B), a gene known to regulate cell cycle and MC hypertrophy.

**Conclusions:** Collectively, the current results demonstrate that up-regulated L13Rik is correlated with DN and may be a hopeful therapeutic target for DN.

## INTRODUCTION

Diabetic nephropathy (DN), occurring in as many as 20–50% of living diabetic patients, is a severe diabetic microvascular complication and a major cause of chronic kidney disease (CKD) and end-stage renal disease worldwide (*Alicic, Rooney & Tuttle, 2017*; *Selby & Taal, 2020*). At the early stage of DN, MC hypertrophy and excessive ECM accumulation are two critical pathological characteristics of CKD (*Rojas-Canales et al., 2019*; *Sun, 2019*). Currently, managing blood sugar and blood pressure and blockading the

renin-angiotensin-aldosterone system (RAAS) are conventional treatments for retarding DN progression (*Chang et al., 2020*; *Choudhury, Tuncel & Levi, 2010*). However, there is no effective treatment to stop or reverse this process. It is essential to reveal the mechanisms by which HG triggers MC hypertrophy and excessive ECM accumulation.

With recent advances in RNA sequencing and bioinformatics algorithms, long non-coding RNAs (lncRNAs) are no longer considered "transcriptional noise" because they do not have protein-coding potential (*Jiang & Zhang, 2021*; *Okazaki et al., 2002*). More and more studies have demonstrated that lncRNA exerts crucial roles in various kinds of biological processes such as genomic imprinting, gene expression, cell proliferation, and tumorigenesis (*Bach & Lee, 2018*; *de Oliveira et al., 2019*; *Sakatani et al., 2005*). As expected, lncRNAs are abnormally expressed in DN patients and mice and exert an important role in the progression of DN (*Coellar, Long & Danesh, 2021*). The data from RNA sequencing show that 106 lncRNAs are dysregulated in kidney tissues of DN mice, and 42 lncRNAs are correlated with DN (*Li et al., 2018*). Functional experiments demonstrated that down-regulated lncRNA-1700020I14Rik attenuates MC proliferation and subsequent ECM accumulation (*Li et al., 2018*). *Zhang et al. (2019)* showed that 95 lncRNAs are dysexpressed in kidney tissues of DN mice compared with normal mice, and up-regulated lncRNA-Rpph1 accelerates MC proliferation and inflammatory response through interacting with the DN-related factor galectin-3. LncRNA-CDKN2B-AS1 expression is increased in the peripheral blood of DN patients and HG-treated MCs, and CDKN2B-AS1 depletion contributes to alleviate MC proliferation and ECM accumulation by interacting with miR-424-5p (*Li et al., 2020*). LncRNA NEAT1 accelerates HG-triggered MC hypertrophy and ECM production by increasing CDKN1B expression (*Liao et al., 2020*). Lnc-MGC inhibits DN progression by regulating MC hypertrophy and ECM production (*Kato et al., 2016*). More and more studies have revealed the important role of lncRNAs and miRNAs in MC hypertrophy (*Dey et al., 2015*, *2012*; *Duan et al., 2017*; *Li et al., 2020*; *Liao et al., 2020*; *Maity et al., 2018*; *Zhang et al., 2014*, *2012*). Given the importance of lncRNAs in DN, it is essential to identify functional lncRNAs during MC injury. L13Rik is an lncRNA that is up-regulated in renal tissues of type 2 diabetes mice compared with normal mice (GSE642 dataset). However, the biological role of L13Rik in regulating MC injury remains unknown.

Cell-cycle arrest is a hallmark of MC hypertrophy (*Shankland, 1999*), and cyclin-dependent kinase inhibitor 1A (CDKN1A), a kind of cyclin-dependent kinase inhibitor, is required for glomerular hypertrophy in experimental DN (*Al-Douahji et al., 1999*). In addition, up-regulation of CDKN1B by lncRNA-NEAT1 promotes MC hypertrophy (*Liao et al., 2020*). In the present study, we demonstrated that L13Rik was significantly increased under HG conditions, and L13Rik depletion effectively reduced HG-induced MC hypertrophy and ECM accumulation through sponging miR-2861 and thus de-repressing its target CDKN1B.

## MATERIALS AND METHODS

Portions of this text were previously published as part of a preprint (*Sun et al., 2022*).

## Patients and samples

Human blood specimens were obtained with the approval of the Ethics Committee of Shanghai Jiao Tong University Affiliated Tong Ren Hospital (No. 2019-060) according to the Declaration of Helsinki. In total, 13 DN blood samples were obtained from DN patients aged 53.3 ± 12.6 (seven men and six women), and 18 control blood samples were obtained from healthy volunteers aged 57.2 ± 11.1 (11 men and seven women) in Tong Ren Hospital. DN was diagnosed in accordance with previous criteria (*An et al., 2015*). Written informed consent was obtained from each donor before sampling. These donors had not received any therapies within 3 months before blood sampling.

## Animal model

All procedures were approved by the Animal Ethics Committee of Shanghai Jiao Tong University Affiliated Tong Ren Hospital (No. 2019-060) and performed in accordance with the ARRIVE guidelines. Male C57BL/6J wild-type mice, aged 8 weeks, were obtained from the Model Animal Research Center of Nanjing University (Nanjing, China) and housed in a thermostatic room (25 ± 2 °C, 12/12 light/dark cycle, 50 ± 10% relative humidity). Animals were allowed to access chow and water *ad libitum*. Mice were randomly divided into a control group ($n = 10$) and a diabetic group ($n = 10$). Experimental DN was induced by intraperitoneal injection with streptozotocin (STZ, 50 mg/kg; Aladdin, Shanghai, China) dissolved in sodium citrate buffer (pH = 4.5). Access to 20% glucose water was allowed *ad libitum* for 18 h to prevent transient hypoglycemia. The blood glucose of each animal was measured from 0 to 35 days after STZ administration. Mice whose blood glucose was greater than 16.7 mmol/L after 3 days of STZ injection and subsequently maintained above this level were considered diabetic (Fig. S1A), as previously described (*Fan et al., 2013*; *Hu et al., 2018*; *Wang et al., 2020*). During the experimental period, any animal suffering from disease or dying was euthanized by cervical dislocation under deep anesthesia using pentobarbital sodium (40 mg/kg).

## Cell culture

A murine mesangial cell line, SV40-MES-13, was purchased from the American Type Culture Collection (ATCC, Natick, MA, USA) and maintained in DMEM (Gibco, New York, CA, USA) containing 10% FBS (Sigma-Aldrich, St. Louis, MO, USA) at 37 °C under 5% $CO_2$. To explore the function of glucose concentration on L13Rik expression, MCs were challenged with 30 mM mannitol (Ma) or glucose at concentrations of 5 (normal glucose, NG), 15, and 30 mM (high glucose, HG). After exposure to glucose, cells were harvested for qRT-PCR and western blot assays.

## Overexpression and RNA interference (RNAi)

The recombinant lentivirus harboring full-length L13Rik (Lv-L13Rik) and negative control lentivirus were purchased from Sangon (Shanghai, China). L13Rik siRNA (siL13Rik) and miR-2861 mimics were synthesized by Genewiz (Jiangsu, China) and transfected into MCs with Lipofectamine RNAiMAX (Invitrogen, Carlsbad, CA, USA) according to the manufacturer's introduction. The sequence for RNAi and miRNA was shown in Table S1.

## Fluorescence *in situ* hybridization (FISH)

The L13Rik probe was synthesized by Sangon (Shanghai, China). After culturing in 12-well plates in the presence or absence of HG for 48 h, SV40-MES-13 cells were washed by PBS and fixed by 4% PFA for 15 min at room temperature, following treatment with proteinase K, glycine and acetic anhydride. Pre-hybridization was performed at 37 °C for 1 h, followed by hybridization of L13Rik against a 250 ng/mL L13Rik probe. Nuclei were labeled with DAPI, and the L13Rik signal was captured as mentioned earlier.

## qRT-PCR

Total RNA in renal tissues and SV40-MES-13 cells was extracted with the RNA Simple Total RNA Kit (Tiangen, Beijing, China). cDNA was synthesized with the TIANScript IIRT Kit (Tiangen, Beijing, China) at 42 °C for 60 min. qRT-PCR was performed in triplicate with FastFire qPCR PreMix (SYBR Green) (Tiangen, Beijing, China) on a ViPlex Fluor real-time PCR system (Vivantis, Selangor, Malaysia). Thermocycling conditions were set as follows: 95 °C for 60 s, followed by 30 cycles of 95 °C for 15 s and 67 °C for 30 s. Gene expressions were calculated by the $2^{-\Delta\Delta CT}$ method. Expressions of fibronectin (FN), collagen IV (Col-IV), N-cadherin (N-cad), CDKN1B, and L13Rik were normalized by GAPDH, and expression of miR-2861 was normalized by U6. All primer sequences were listed in Table S1.

## Western blot

SV40-MES-13 cells were lysed with the xTractor Buffer Kit (Clontech Laboratories, Inc., Mountain View, CA, USA). Protein specimens were quantified through the BCA assay kit (Beyotime, Shanghai, China) and electrophoresed through 10% SDS-PAGE gels. After transferring onto PVDF membranes, interested proteins were masked with 2% non-fat milk at 37 °C for 30 min, then incubated with primary antibodies against FN (1:700, SAB5700724; Sigma-Aldrich, St. Louis, MO, USA), Col-IV (1:500, ab52235; Abcam, Fremont, CA, USA), CDKN1B (1:1,000, ab32034; Abcam, Fremont, CA, USA), N-cad (1:8,000, ab76011; Abcam, Fremont, CA, USA), Bax (1:3,000, ab32503; Abcam, Fremont, CA, USA), Bcl-2 (1:1,500, ab196495; Abcam, Fremont, CA, USA), and β-actin (1:20,000, AF7018; Affinity, Jiangsu, China) at 4 °C for 20 h. The membranes were rinsed three times with PBST, then immersed in HRP-conjugated anti-rabbit IgG secondary antibody solution (1:5,000, S0001; Affinity, Jiangsu, China) at 37 °C for 60 min. Interesting proteins were visualized using a commercial ECL kit (Glpbio, Montclair, CA, USA). Fluorescence intensities were quantified using Fiji software.

## RNA pull-down assay

Biotin-labeled miR-2861 was purchased from Genewiz (Jiangsu, China). Biotin-labeled miR-2861 (3 μg) was incubated with streptavidin-coated beads (434341; Thermo Fisher Scientific, Waltham, MA, USA) at 4 °C overnight. Then, streptavidin-coated beads were incubated with SV40-MES-13 cell lysate at 4 °C for 12 h. RNAs bound by miR-2861 were purified with Trizol reagent (Takara, Shiga, Japan) and assessed through qRT-PCR.

## RNA immunoprecipitation (RIP) assay

After miR-2861 knockdown, the RIP assay was performed as per the instructions of the Immunoprecipitation Kit with Protein A+G Magnetic Beads (Beyotime, Shanghai, China). At first, $1 \times 10^6$ SV40-MES-13 cells were lysed with 200 μL of RIPA buffer (Beyotime, Shanghai, China). Lysates were incubated with 20 μL of magnetic beads labeled with normal mouse IgG (A7028; Beyotime, Shanghai, China) or rabbit anti-Ago2 antibody (DF12246; Affinity, Jiangsu, China) at 4 °C for 16 h. Immunoprecipitated RNA was obtained by digesting proteins with Proteinase-K, and quantified on a microplate reader. Lastly, immunoprecipitated L13Rik and miR-2861 were assessed through qRT-PCR.

## Dual-luciferase reporter assay

The wild-type or mutant predictive binding site of miR-2861 on L13Rik was cloned into the PGL3 vector (Fenghui Biotechnology, Hunan, China) to construct pGL3-L13Rik-WT or pGL3-L13Rik-Mut plasmids. The 3′-UTR of CDKN1B or its mutant was inserted into the PGL3 vector to construct pGL3-CDKN1B-3′UTR-WT or pGL3-CDKN1B-3′UTR-Mut plasmids. HEK293 cells ($1 \times 10^6$) were plated into 24-well plates. pGL3-L13Rik-WT (20 ng), pGL3-L13Rik-Mut (20 ng), pGL3-CDKN1B-3′UTR-WT (20 ng), or pGL3-CDKN1B-3′UTR-Mut (20 ng) plasmids were co-transfected with 40 nM of miR-2861 and 2 ng of pRL-TK (Beyotime, Shanghai, China) into HEK293 cells with Lipofectamine™ 3,000 according to the manufacturer's protocol. After incubation at 37 °C for 48 h, luciferase activities were assessed using the Dual-Luciferase Reporter Gene Assay Kit (Beyotime, Shanghai, China) in accordance with the manufacturer's protocol. Renilla luciferase was used for internal control.

## Protein synthesis

After treatment with HG, SV40-MES-13 cells were incubated with [$^{35}$S]-methionine and then protein synthesis was assessed as described previously (*Mahimainathan et al., 2006*).

## Cell hypertrophy

SV40-MES-13 cells were treated with HG and then lysed with the xTractor Buffer Kit. Protein content was quantified through the BCA assay kit, and cell hypertrophy was evaluated by testing the ratio of total protein to cell number (*Mahimainathan et al., 2006*).

## Flow cytometry

SV40-MES-13 cells were treated with HG and dyed with propidium iodide (PI) for 1 h. the cell cycle was measured using an Attune N×T flow cytometer (Thermo Fisher Scientific, Waltham, MA, USA).

## Statistical analysis

Data were presented as mean ± standard deviation (SD) from three independent experiments and analyzed using SPSS 22.0 software (IBM, Armonk, NY, USA).
The difference among groups was compared with the student's *t*-test or one-way analysis of variance (ANOVA) followed by the Scheffé test. A *p*-value less than 0.05 was defined as statistically significant.

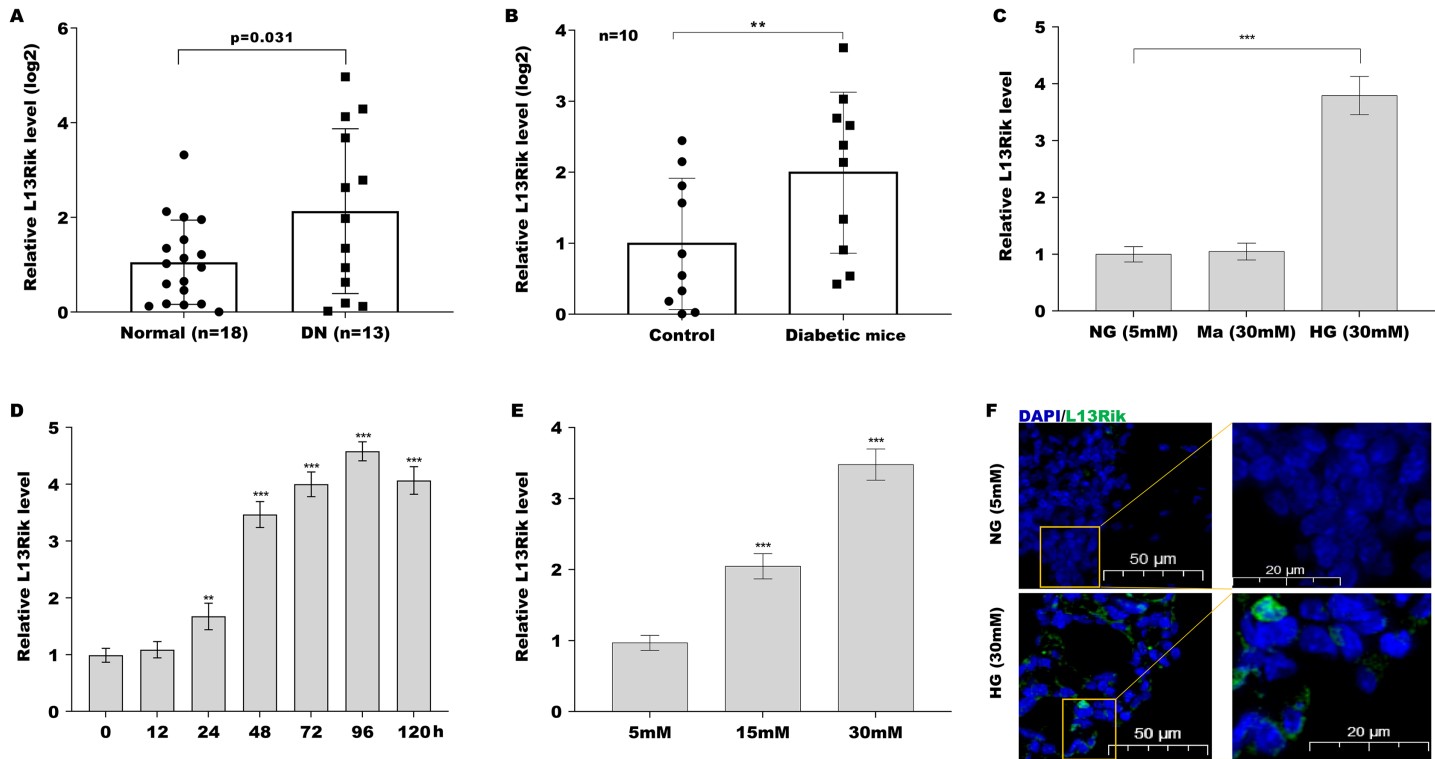

**Figure 1  L13Rik was up-regulated in DN patients, diabetic mice, and HG-treated SV40-MES-13 cells.** $**p < 0.01$, $***p < 0.001$.

## RESULTS

### L13Rik was increased in DN patients, diabetic mice, and HG-treated MCs

Given that L13Rik expression is up-regulated in renal tissues of mice with type two diabetes compared with normal mice (Figs. S1B and S1C), we next investigated whether L13Rik is correlated with DN progression through regulating MC hypertrophy. To this end, L13Rik levels were first assessed in DN patients and diabetic mice. As shown in Fig. 1A, L13Rik levels were significantly up-regulated in the peripheral blood of DN patients ($n = 13$) compared with healthy controls ($n = 18$). Similarly, L13Rik expression was also increased in the renal tissues of diabetic mice compared with control mice (Fig. 1B). Consistently, L13Rik expression was remarkably up-regulated in HG-cultured SV40-MES-13 cells (Fig. 1C). The relationships between treatment time and glucose concentration on L13Rik expression were also explored. As shown in Figs. 1D and 1E, HG treatment resulted in a significant increase in the L13Rik level in a time- and dose-dependent manner. Furthermore, the results from the FISH assay verified that L13Rik was up-regulated in HG-cultured SV40-MES-13 cells and that L13Rik was mainly located in the cytoplasm of SV40-MES-13 cells (Fig. 1F).

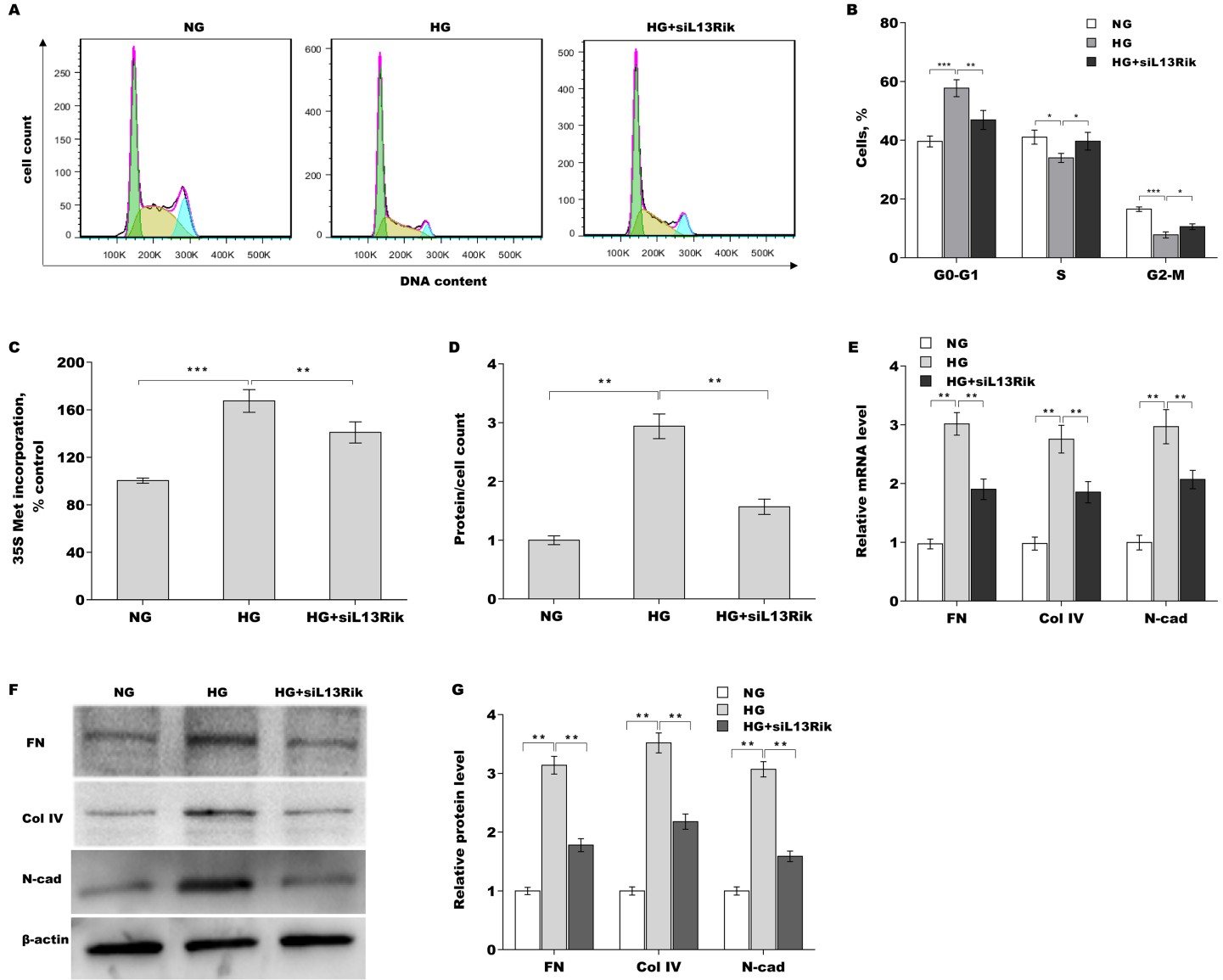

**Figure 2** L13Rik depletion alleviated HG-induced MC hypertrophy and ECM accumulation. $^*p < 0.05$, $^{**}p < 0.01$, $^{***}p < 0.001$.

## L13Rik depletion alleviated HG-induced MC hypertrophy and ECM accumulation

Cell cycle arrest in the G1 phase and subsequent hypertrophy are critical pathological characteristics of MC injury (*Rojas-Canales et al., 2019*; *Sun, 2019*). To investigate the biological effect of L13Rik on regulating MC injury, SV40-MES-13 cells were treated with HG alone or combined with siRNA against L13Rik (siL13Rik), and then the cell cycle was assessed through flow cytometry. Figures 2A and 2B showed that HG treatment resulted in the arrest of cells at the G1 phase, whereas the changes were reversed by L13Rik knockdown. The protein levels were next examined in SV40-MES-13 cells after treatment with HG and siL13Rik. HG enhanced the amount of newly synthesized proteins, as

evidenced by $^{35}$S-methionine incorporation (Fig. 2C). HG further resulted in cell hypertrophy, as suggested by elevated total protein content in SV40-MES-13 cells (Fig. 2D). These effects were blocked by L13Rik knockdown (Figs. 2C and 2D).

To identify the effect of L13Rik on ECM accumulation in SV40-MES-13 cells, FN, Col-IV, and N-cad expression were assessed by qRT-PCR and western blot analysis after treatment with HG in the presence or absence of siL13Rik. Figures 2E–2G showed that the mRNA and protein levels of FN, Col-IV, and N-cad were markedly increased after treatment with HG, whereas these effects were blocked by L13Rik depletion.

## L13Rik acted as a ceRNA to sponge miR-2861

LncRNA, located in the cytoplasm, commonly acts as ceRNA to control gene expression *via* sponging specific miRNAs (*Thomson & Dinger, 2016*). Given that L13Rik mainly locates in the cytoplasm in SV40-MES-13 cells (Fig. 1F), we speculated that L13Rik might exert its biological effect in this way. To prove that, a bioinformatics tool, miRDB (http://mirdb.org/cgi-bin/search.cgi) (*Chen & Wang, 2020*), was applied to predict miRNAs sponged by L13Rik, and 63 candidate miRNAs were identified (Table S2). A previous study demonstrated that 86 miRNAs were dysexpressed in DN progression (Table S3) (*Chen et al., 2020*). Venn diagram analysis revealed that miR-2861 was the only miRNA that could be sponged by L13Rik and dysregulated in DN patients (Fig. S3).

The miR-2861 level was next assessed in DN patients and diabetic mice. As shown in Fig. 3A, miR-2861 expression was obviously decreased in the peripheral blood of DN patients ($n = 13$) compared to healthy controls ($n = 18$). miR-2861 was also down-regulated in the renal tissues of diabetic mice compared to control mice (Fig. 3B). In addition, HG treatment resulted in a significant decrease in miR-2861 in a dose-dependent manner (Fig. 3C). To verify the direct combination of miR-2861 with L13Rik, the binding sites between L13Rik and miR-2861 were analysed, and recombinant plasmids containing the binding sites (pGL3-L13Rik-Wt and pGL3-L13Rik-Mut) were constructed (Fig. 3D). pGL3-L13Rik-Wt or pGL3-L13Rik-Mut was co-transfected with miR-2861 into HEK293 cells, and then relative luciferase activity was measured using the dual luciferase reporter assay. As shown in Fig. 3E, miR-2861 significantly decreased the luciferase activity of pGL3-L13Rik-Wt, but did not affect the luciferase activity of pGL3-L13Rik-Mut. Furthermore, the miR-2861 mutant lost the inhibitor effect on the luciferase activity of pGL3-L13Rik-Wt (Fig. 3F), indicating that the combination of miR-2861 with L13Rik was sequence-specific. The results from the RNA pull-down assay further demonstrated that L13Rik was more enriched in the biotin-labeled miR-2861 precipitates than in the biotin-labeled miR-cont precipitates (Fig. 3G). Argonaute 2 (Ago2) is the core component of the "miRNA-induced silencing complex (miRISC)", a multi-protein complex that incorporates miRNA and its target mRNA (or lncRNA) (*Chendrimada et al., 2005*; *Liu et al., 2004*). The results from RIP with Ago2 antibody showed that miRNA-2861 and L13Rik were concurrently enriched in SV40-MES-13 cells, and miRNA-2861 inhibitor significantly reduced miRNA-2861 level and increased L13Rik enrichment in Ago2 precipitates (Figs. 3H and 3I). These results demonstrate that L13Rik functions as a ceRNA to sponge miR-2861 in a sequence-specific manner.

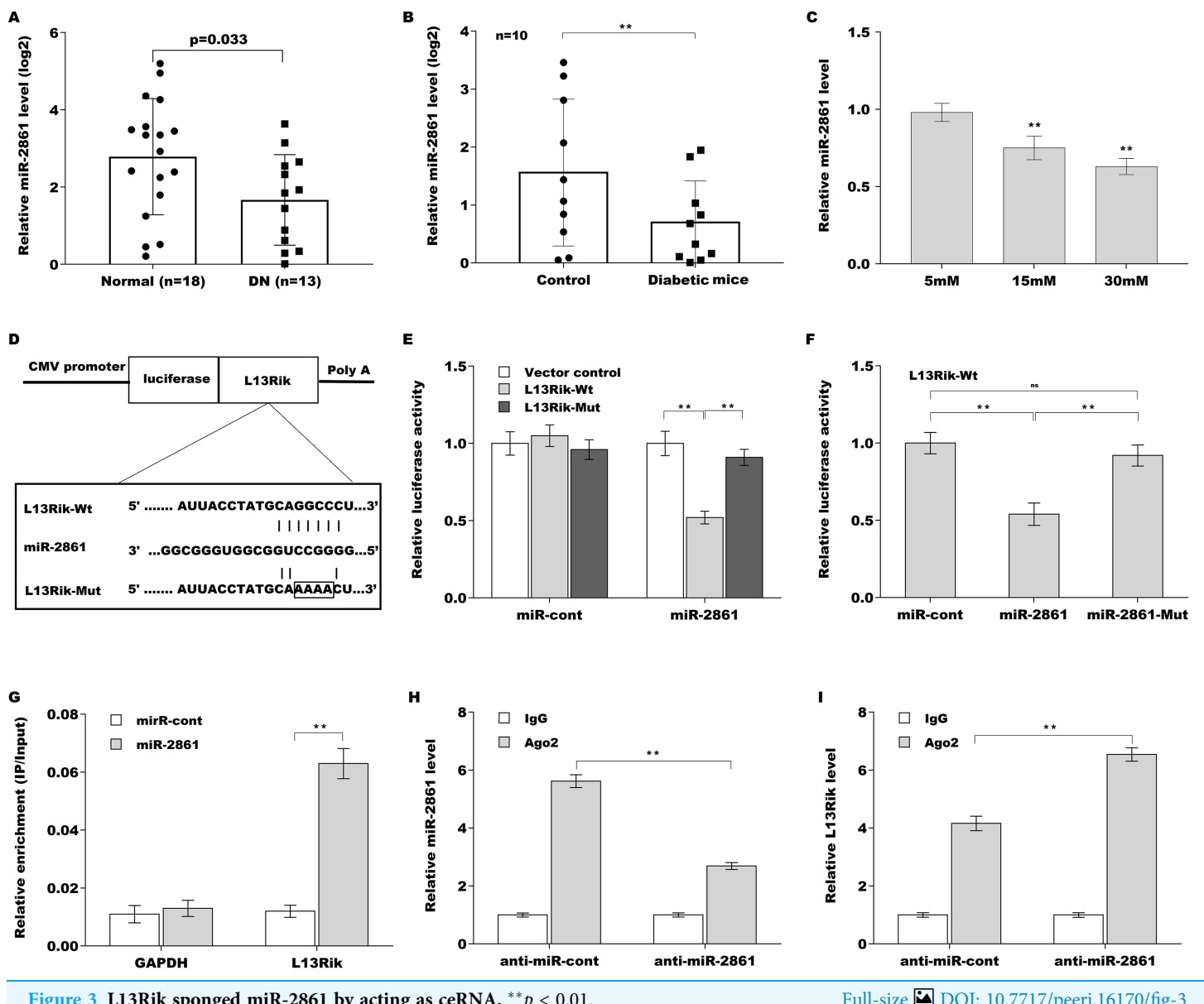

**Figure 3 L13Rik sponged miR-2861 by acting as ceRNA.** $^{**}p < 0.01$.

## MiR-2861 suppressed HG-induced MC hypertrophy and ECM accumulation

The roles of miR-2861 in regulating MC hypertrophy and ECM accumulation were next investigated. As shown in Figs. 4A and 4B, miR-2861 overexpression alleviated HG-induced arrest of cells at the G1 phase. miR-2861 further reduced HG-induced new protein synthesis (Fig. 4C) and cell hypertrophy (Fig. 4D), as evidenced by decreased $^{35}$S-methionine incorporation and total protein content, respectively. To identify the effect of miR-2861 on ECM accumulation in SV40-MES-13 cells, FN, Col-IV, and N-cad expression were assessed by qRT-PCR and western blot analysis after treatment with HG in the presence or absence of miR-2861. Figure 4E showed that miR-2861 significantly repressed the HG-induced increase in FN, Col-IV, and N-cad mRNA levels. miR-2861 also

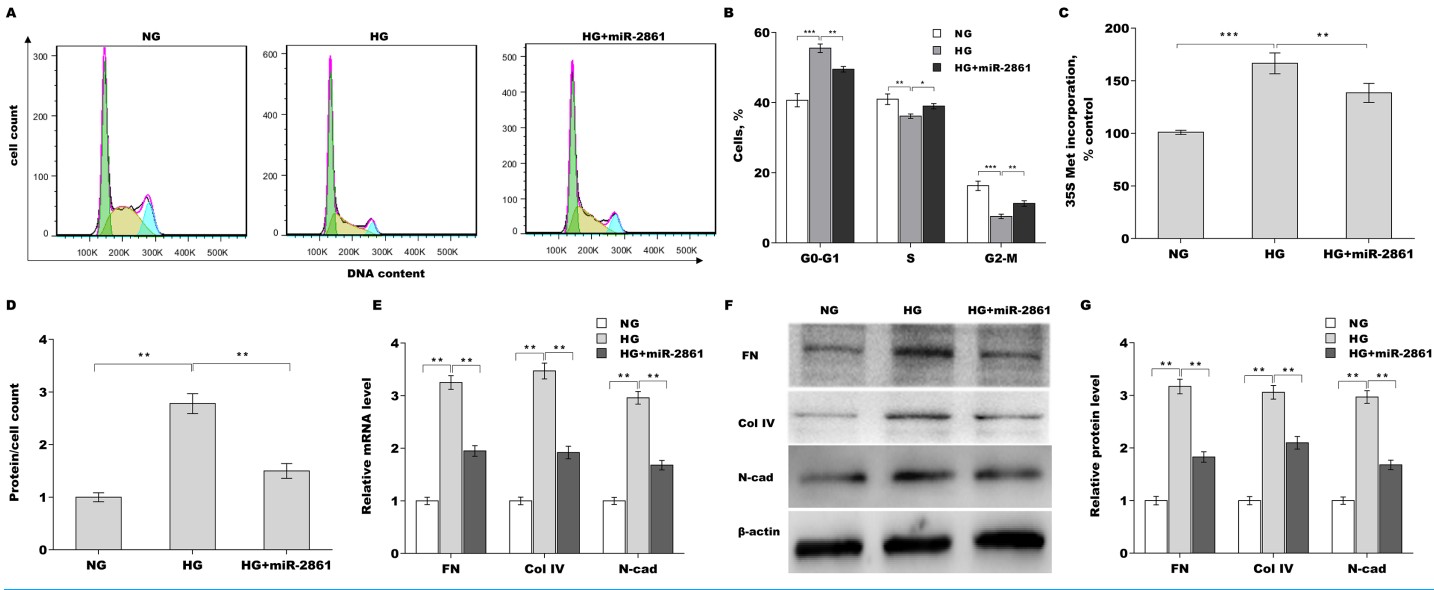

**Figure 4** MiR-2861 suppressed HG-induced MC hypertrophy and ECM accumulation. $**p < 0.01$, $***p < 0.001$.

significantly suppressed the HG-induced increase in FN, Col-IV, and N-cad protein levels (Figs. 4F and 4G).

## MiR-2861 antagonized the effect of L13Rik on MC hypertrophy

To explore whether L13Rik accelerated MC hypertrophy through sponging miR-2861, SV40-MES-13 cells were overexpressed by L13Rik in the presence or absence of miR-2861 mimic, and cell hypertrophy and ECM accumulation were assessed. L13Rik overexpression resulted in the arrest of cells at the G1 phase (Fig. 5A), enhanced the amount of newly synthesized proteins (Fig. 5B), and caused cell hypertrophy (Fig. 5C). All these effects were reversed by miR-2861 (Figs. 5A–5C). Moreover, L13Rik increased the mRNA and protein expression of FN, Col-IV, and N-cad, whereas miR-2861 overexpression reversed these changes (Figs. 5D–5F).

## L13Rik increased CDKN1B expression by sponging miR-2861

The potential target genes of miR-2861 were predicted using the TargetScan tool (http://www.targetscan.org/vert_71/). A total of 4,407 genes were predicted to be potential targets of miR-2861. Among these genes, CDKN1B was reported to be associated with cell cycle and MC hypertrophy (*Awazu et al., 2003*; *Liao et al., 2020*). To reveal the regulatory role of miR-2861 in CDKN1B expression, recombinant plasmids of pGL3-CDKN1B-3′UTR-Wt and pGL3-CDKN1B-3′UTR-Mut were constructed and co-transfected with miR-2861 (Fig. 6A). As shown in Fig. 6B, miR-2861 significantly decreased the luciferase activity of pGL3-CDKN1B-3′UTR-Wt but had no impact on the luciferase activity of pGL3-CDKN1B-3′UTR-Mut. Moreover, CDKN1B protein expression was further assessed in SV40-MES-13 cells after treatment with the miR-2861 mimic. Figures 6C and 6D showed that CDKN1B expression was obviously decreased after miR-2861 overexpression. More

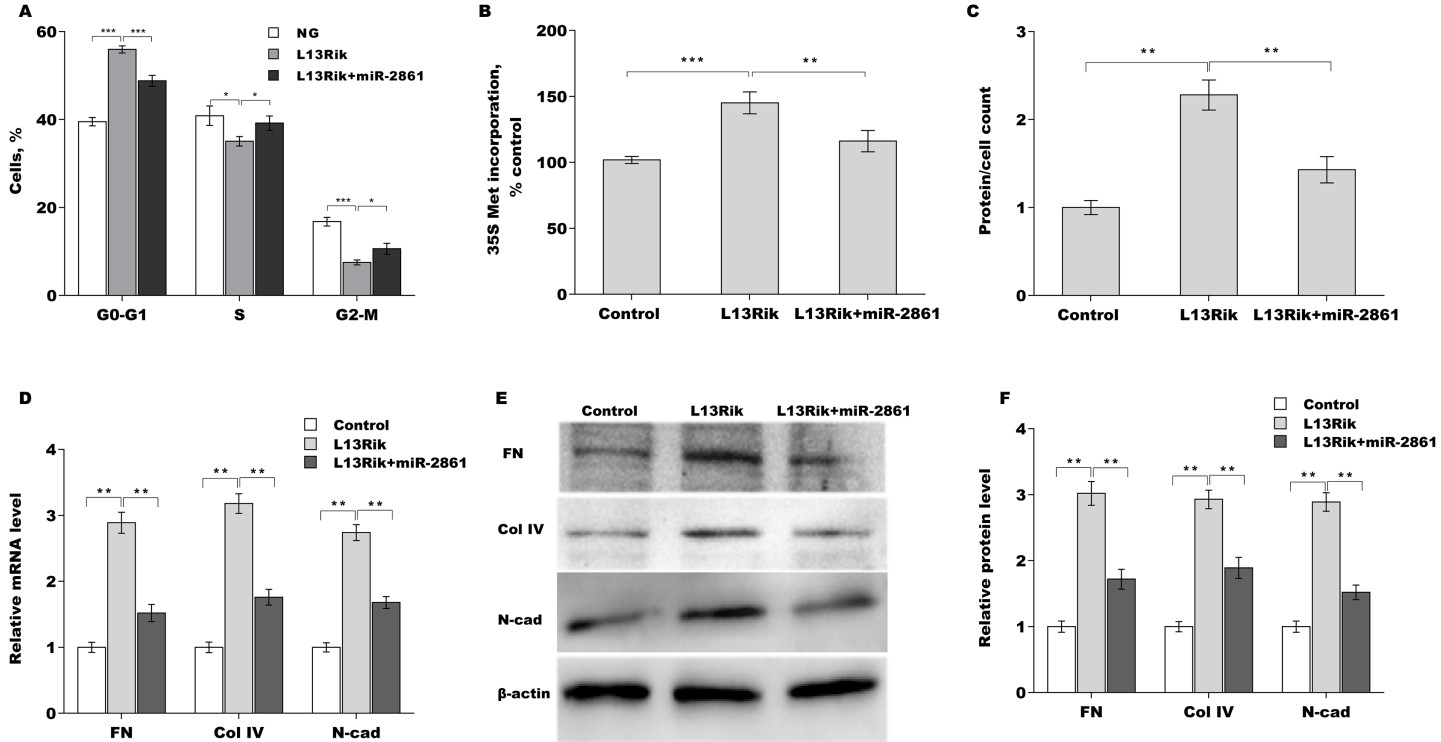

**Figure 5  MiR-2861 antagonized the effect of L13Rik on MC hypertrophy.** $^*p < 0.05$, $^{**}p < 0.01$, $^{***}p < 0.001$.

importantly, L13Rik increased CDKN1B expression in SV40-MES-13 cells, whereas miR-2861 overexpression reversed the effect (Figs. 6E and 6F).

## DISCUSSION

MC injury, including hypertrophy and excessive ECM production, plays a crucial role in the pathology of DN (*Dai, Liu & Liu, 2017*). LncRNA was reported to be involved in MC injury and DN progression (*Coellar, Long & Danesh, 2021*). However, the role of lncRNA in these processes is still unclear. In the current study, we demonstrated that L13Rik mediates HG-induced MC hypertrophy and ECM accumulation by regulating the miR-2861/CDKN1B axis, as evidenced by the following: (i) L13Rik was increased in DN patients, diabetic mice, and HG-treated MCs; (ii) L13Rik depletion alleviated HG-induced MC hypertrophy and ECM accumulation; (iii) L13Rik acted as ceRNA to sponge miR-2861; (iv) MiR-2861 suppressed HG-induced MC hypertrophy and ECM accumulation; (v) MiR-2861 antagonized the effect of L13Rik on MC hypertrophy; (vi) L13Rik increased CDKN1B expression by sponging miR-2861. These data revealed the function and underlying mechanism of L13Rik in regulating MC hypertrophy and may provide a potential opportunity to treat DN.

L13Rik is an lncRNA of 1,726 bases in length that is located on chromosome 3 (Chr3: 95,871,521–95,889,093). The abnormal expression of L13Rik has been revealed in HG- or TGF-β-treated MCs (GSE2557 and GSE2558) and renal tissues from type two diabetic mice (GSE642) using RNA sequencing and microarray analysis. However, the biological

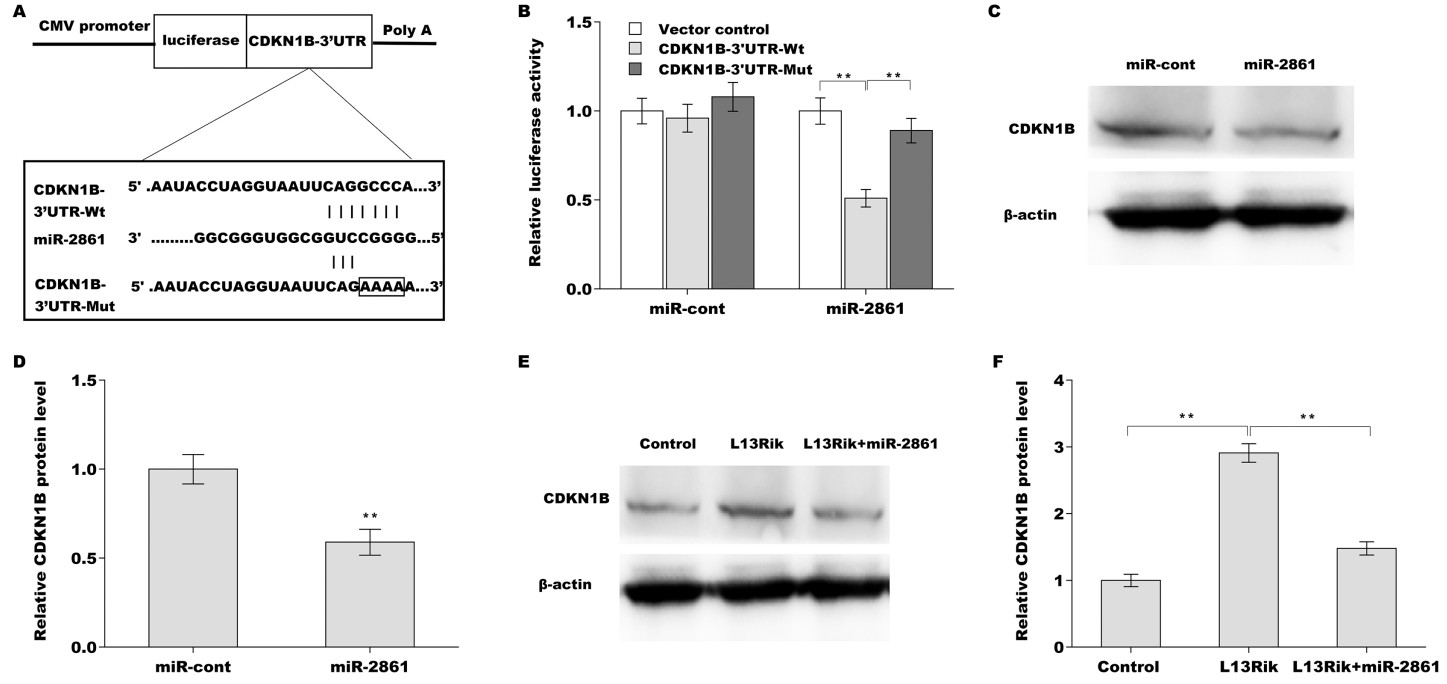

**Figure 6 L13Rik promoted MC hypertrophy by modulating miR-2861/CDKN1B axis.** $^{**}p < 0.01$.

role of L13Rik in DN progression remains unknown. In the study, we demonstrated that up-regulated L13Rik exerts a significant role in regulating HG-triggered cell hypertrophy and EMC accumulation. L13Rik depletion remarkably alleviates HG-induced MC injury. Given that the kidney is mainly composed of three types of cells: MC, podocyte, and endothelial cells (*Wallace, 1998*), it is essential to further explore whether L13Rik regulates HG-induced podocyte and endothelial cell injury. In addition, the effect of L13Rik on DN progression will be investigated in diabetic mice.

Mounting studies have demonstrated that lncRNA functions as an important mediator in DN or other diabetic complications (*Lu et al., 2021*). For instance, LncRNA NEAT1 expression is increased in DN patients and HG-treated MCs, and NEAT1 over-expression accelerates MC hypertrophy through sponging miR-222-3p and thus increasing CDKN1B expression (*Liao et al., 2020*). As a kind of decoy that competes for miRNAs, ceRNA incorporates non-coding RNA (miRNA and lncRNA) with mRNAs in miRISC through miRNA response elements (MREs) (*Karreth & Pandolfi, 2013*). According to this hypothesis, lncRNAs containing MREs can segregate miRNAs from mRNAs containing the same MREs, derepressing the mRNA expression (*Guo et al., 2010*; *Karreth & Pandolfi, 2013*). In fact, more and more studies have revealed that lncRNAs function in this way. In the study, we demonstrated that L13Rik acts as a ceRNA to sponge miR-2861, resulting in the de-repression of its target CDKN1B, a gene known to accelerate MC hypertrophy (*Liao et al., 2020*).

A previous study found that miR-2861 is decreased in DN patients and associated with an estimated glomerular filtration rate (*Cardenas-Gonzalez et al., 2017*). However, the

mechanism of miR-2861 in the development of DN is still unclear. The current results revealed that miR-2861 is decreased in DN mice and HG-induced MCs. Functionally, miR-2861 overexpression represses HG-induced MC hypertrophy and ECM accumulation.

## CONCLUSIONS

L13Rik accelerates HG-induced mesangial cell hypertrophy by regulating the miR-2861/CDKN1B axis.

### Funding

This work was funded by the Natural Science Foundation of Shanghai (Grant Number 20ZR1451600) and the Shanghai Municipal Health Bureau Project (Grant Number 201940439). The funders had no role in study design, data collection and analysis, decision to publish, or preparation of the manuscript.

### Grant Disclosures

The following grant information was disclosed by the authors:
Natural Science Foundation of Shanghai: 20ZR1451600.
Shanghai Municipal Health Bureau Project: 201940439.

### Competing Interests

The authors declare that they have no competing interests.

### Author Contributions

- Linlin Sun conceived and designed the experiments, performed the experiments, authored or reviewed drafts of the article, and approved the final draft.
- Miao Ding performed the experiments, analyzed the data, authored or reviewed drafts of the article, and approved the final draft.
- Fuhua Chen performed the experiments, analyzed the data, authored or reviewed drafts of the article, and approved the final draft.
- Dingyu Zhu performed the experiments, analyzed the data, prepared figures and/or tables, and approved the final draft.
- Xinmiao Xie performed the experiments, prepared figures and/or tables, and approved the final draft.

### Human Ethics

The following information was supplied relating to ethical approvals (*i.e.*, approving body and any reference numbers):

Shanghai Jiao Tong University Affiliated Tong Ren Hospital granted Ethical approval to carry out the study within its facilities (Ehtical Application Ref: 2019-060).

## Data Availability

The raw measurements are available in the Supplemental Files.

## Supplemental Information

Supplemental information for this article can be found online at http://dx.doi.org/10.7717/peerj.16170#supplemental-information.

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
