# Peer review of "Long non‑coding RNA L13Rik promotes high glucose-induced mesangial cell hypertrophy and matrix protein expression by regulating miR-2861/CDKN1B axis"

_PeerJ, doi:10.7717/peerj.16170_

## Round 0.1 · original submission · Major Revisions

Something of a split decision by the two reviewers here, hence I am giving you a chance to rebut. However, on balance I share many of the concerns of reviewer-1 and wish to be clear that I consider extra work to be essential, and a careful re-writing of the manuscript is essential.

I need you to carefully address the concerns of reviewer-1 regarding the STZ-induction of diabetes. 72h is in my view too short a time for clear induction of DN and I share the concern that the expression of L13Rik may be induced by STZ-injection but not by diabetes. Physiological (blood glucose levels, proteinuria and etc) and histological (PAS staining, ECM accumulation, mesangial hypertrophy and etc) data of those diabetic rats must be presented.

A power calculation for the human studies is essential.

A full set of uncropped blots for all technical and biological replicates must be provided in supplemental images.

Clarity of statistical tests needs to be provided. I do not find the human data compelling.

Evidence of MC injury must be provided.

Detail the sex of the participants and rodents used.

I note also that reviewer-2 has asked for additional data - this must also be provided.

Reviewer 1 ·

Basic reporting

The authors found the higher expression of lncRNA C920021L13Rik (L13Rik) in blood from human DN patients, kidneys from diabetic rats and mesangial cells treated with high glucose. siRNA against L13Rik reversed the effects of HG in cell cycle arrest, protein levels and ECM gene expression in MC. They also found miR-2861 targets L13Rik and antagonizes the effects of L13Rik in cell cycle arrest, protein levels and ECM gene expression. miR-2861 also targets CDKN1B and the effects were reversed by L13Rik. Those results suggest that L13Rik may reverse the effects of miR-2861 on cell cycle arrest in diabetic conditions.
However, the current manuscript has several major concerns.

Especially it is not clear enough that kidney injury and hypertrophy are regulated by L13Rik.

Many important literatures related to glomerular mesangial cell hypertrophy in diabetes and noncoding RNAs (lncRNA and miRNAs) are missing in references and this manuscript is quite biased.

The reason why the authors chose L13Rik is not clear.

Authors called it a “novel” L13Rik RNA. What do the authors mean “novel”? Nobody identified it before? The authors are the first to find the transcript?

Experimental design

If some other lncRNAs are up-regulated in the same samples, include the data of other lncRNAs as positive control. It is better to include some negative controls (lncRNAs) which never respond to diabetic conditions.

Table 1 shows PCR primers. Authors used human blood, rat kidney and mouse mesangial cells. Sequences of many genes (including noncoding RNAs) are not 100% conserved from mouse to human. One PCR primer pair can’t be useful for all of the species. What species does Table 1 show?

Sample numbers of human blood and diabetic rats are not enough for statistics.

Diabetes was induced by STZ injection in rats. Usually DN is not induced in 72 hours of diabetes. The expression of L13Rik may be induced by STZ-injection but not by diabetes. Physiological (blood glucose levels, proteinuria and etc) and Histological (PAS staining, ECM accumulation, mesangial hypertrophy and etc) data of those diabetic rats should be shown.

Validity of the findings

Fig.3 is the evidence that miR-2861 regulates (targets) L13Rik but not that L13Rik is ceRNA (sponge) of miR-2861.

Fig.5&6. No evidence of MC injury. Authors just showed the expression of ECM (not the evidence of injury). Actual kidney injury (with protection by miR-2861 or promotion by L13Rik) in diabetic rats should be shown.

CDKN1B is cell cycle regulator but not a regulator of kidney injury or hypertrophy. Some direct targets of miR-2861 related to hypertrophy (protein synthesis or degradation) should be described.

Bar graphs
Fig.1ABFig.3AB
Sample sides are small and each group has big deviation. Statistics should be confirmed (the differences are really statistically significant?).The authors need more samples to verify the conclusion.

Many bar graphs look similar and similar SD (or SE?).
What is shown in bar graphs? Mean and SE or SD? Sample sides? What statistics? They should be shown in figure legends.

Western Blots are not clear.
Molecular weight markers should be shown.
Wider scans with molecular weight markers should be shown in supplementary info.

·

Basic reporting

The manuscript is well written with a clear background and problem statement and how this research is addressing a gap in the field. The figures are clear and easy to follow. Overall, the authors did a commendable job presenting the data very clearly and effectively.

A few minor comments on language noted below:
• Line 34: Suggest changing to: “Diabetic nephropathy (DN), occurring in as many as 20-50% of living diabetic patients, is a severe…”
• Line 39: conventional “treatments”
• Line 41: “stop or reverse this process”
• Line 46-47: Unclear how gene imprinting, gene expression are pathophysiological processes. Suggest author to clarify.
• Line 60-61: Suggest changing to: “L13Rik is a lncRNA that is upregulated in renal tissues of type 2 diabetes mice compared with normal mice…”
• Line 64: “Cell cycle arrest is…” (remove The)
• Line 68: “In the present study, we demonstrated 68 that L13Rik was significantly increased…”, please clarify it is increased under HG conditions.
• Line 222: compared “to” control rats

Experimental design

The research question is well defined and has been succinctly presented in the background. The authors have performed all the experiments thoroughly with the right controls which provide confidence to the data. Through knockdowns and over-expressions, the authors have convincingly shown the regulation that L13Rik acts as a ceRNA to sponge miR-2861, thus derepressing CDKN1B. The methods are clearly written as well.

I suggest the authors address these following points that will further strengthen their data:
• Line 183: Figure S1 shows L13Rik expression in non-diabetic and diabetic renal tissue. Can the author quantify the level of over-expression in diabetic tissue with some statistical analysis and mention that in the main text
• Figure 1D: Did the authors look at when L13Rik expression goes down post HG treatment? The kinetics of expression would be a valuable addition to the manuscript to understand how sustained the L13Rik expressions are in response to HG treatment
• Figure 3F: Also suggest showing stats between miR-cont and miR-2861-mut

Validity of the findings

The data presented here is very robust and backed by well-controlled and well-designed experiments. Having looked at the regulation of L13Rik and miR-2831 in three different models of diabetes including DN patient, diabetic rats and HG exposed HG-cultured SV40-MES-13 cells further strengthen the quality of the manuscript and validity of the findings. The authors also present the conclusions of the data succinctly and within the bounds of what the data shows.

Additional comments

The authors have done a fantastic job with the research as well as presenting it in this manuscript. I believe this will be of great interest to the scientific community.

---

## Round 0.2 · Minor Revisions

Nearly there - please attend to the minor requests indicated.

Thank you for attending to these matters.

Reviewer 1 ·

Basic reporting

The revised manuscript has been improved.

To avoid unnecessary troubles after publication, I recommend a few more things below.

Western blots.
Authors responded like below. But I don’t see any molecular weight markers in the figures. Put also molecular weights in the marker lane and sample names in the lanes in supplementary. And confirm again which parts (in supplements) are used for main figures.

Response: Molecular weight markers have be shown in the revised Figures.
A full set of uncropped blots for three biological replicates have been provided in supplemental images.

I see typos and grammatical errors even in the Response. I recommend checking such errors before publication.

I recommend confirming if references are properly cited and described correctly again.

Experimental design

no

Validity of the findings

no

Additional comments

no

·

Basic reporting

Authors have addressed feedback reported in this section.

Experimental design

Authors have addressed feedback reported in this section.

Validity of the findings

Authors have addressed feedback reported in this section.

Additional comments

-

---

## Round 0.3 · accepted · Accept

Thanks for addressing the final issues.